# Stochastic Analysis of Hourly to Monthly Potential Evapotranspiration with a Focus on the Long-Range Dependence and Application with Reanalysis and Ground-Station Data

**Panayiotis Dimitriadis** [ID]**, Aristoteles Tegos * and Demetris Koutsoyiannis** [ID]

Department of Water Resources and Environmental Engineering, School of Civil Engineering, National Technical University of Athens, Heroon Polytechneiou 5, 15880 Zographou, Greece; pandim@itia.ntua.gr (P.D.); dk@itia.ntua.gr (D.K.)
* Correspondence: tegosaris@yahoo.gr

**Abstract:** The stochastic structures of potential evaporation and evapotranspiration (PEV and PET or ETo) are analyzed using the ERA5 hourly reanalysis data and the Penman–Monteith model applied to the well-known CIMIS network. The latter includes high-quality ground meteorological samples with long lengths and simultaneous measurements of monthly incoming shortwave radiation, temperature, relative humidity, and wind speed. It is found that both the PEV and PET processes exhibit a moderate long-range dependence structure with a Hurst parameter of 0.64 and 0.69, respectively. Additionally, it is noted that their marginal structures are found to be light-tailed when estimated through the Pareto–Burr–Feller distribution function. Both results are consistent with the global-scale hydrological-cycle path, determined by all the above variables and rainfall, in terms of the marginal and dependence structures. Finally, it is discussed how the existence of, even moderate, long-range dependence can increase the variability and uncertainty of both processes and, thus, limit their predictability.

**Keywords:** potential evapotranspiration; stochastic simulation; marginal structure; long-range dependence; Hurst–Kolmogorov dynamics

## 1. Introduction

Evapotranspiration is a paramount element in hydrology, with relevance in many aspects of the geosciences. From hydrological and agronomic perspectives, the potential evapotranspiration (PET) and (potential) evaporation (PEV) are key for water balance estimation, the assessment of crop water demand, and integrated rainfall-runoff modelling. PET [1] is defined as "the amount of water transpired in a given time by a short green crop, completely shading the ground, of uniform height and with adequate water status in the soil profile". A particular (reference) case thereof is the reference evapotranspiration (ETo), which refers to "the rate of evapotranspiration from a hypothetical reference crop with an assumed crop height of 0.12 m, a fixed surface resistance of 70 s/m, and an albedo of 0.23, closely resembling the evapotranspiration from an extensive surface of green (cool season) grass of uniform height, actively growing, well-watered, and completely shading the ground" [2]. Evaporation is the physical process by which liquid water enters the atmosphere as water vapor. In what follows, when we refer to all of the above processes, we use the acronym PE. We also note that PE may be different from the actual evapo(transpi)ration (in cases where there is not adequate water availability).

For the PE assessment, historically, many models have been developed highlighting the Penman–Monteith model as the most suitable [3]. One of the main shortcomings of estimating PE with the Penman–Monteith model is the requirement of a significant number of meteorological inputs such as, without distinction, temperature, radiation, relative

humidity, and wind speed [4–6]. In the case that we require a synthetic PE timeseries for risk management (e.g., in a Monte Carlo simulation framework), when the above meteorological inputs are not available for the requested period, then one may use a stochastic model that preserves the important statistical attributes of PE. Additionally, due to the physical complexity of assessing the PE, stochastic modelling provides a solid scientific ground for further consideration in several fields of PE assessment, and the stochastic analysis can contribute to the PE physical interpretation along with other hydrometeorological processes because stochastics is proven as a collection of mathematical tools able to give physical explanations [7]. As highlighted in the aforementioned work [7], the role of stochastics is crucial: (a) to infer dynamics (laws) from past data; (b) to formulate the complex natural system equations; (c) to estimate the involved parameters; and (d) to test any hypothesis regarding the dynamics. There are only limited works providing a thorough stochastic analysis in PE timeseries, even though the necessity of stochastic modelling is of paramount importance. Based on the published literature, a seasonal ARIMA model and Winters' exponential smoothing model [8] have been investigated for their applicability for forecasting weekly reference crop ETo [9]. Both models demonstrated satisfactory results compared to a simple PE model. Pandey et al. [10] provided a stochastic analysis in assessing black gram evapotranspiration regimes using a long-term pan-evaporation dataset of 23 years in Udaipur, India. Black gram is an important crop of the Udaipur region, and the lack of long-term crop demand assessment led to the need for stochastic analysis using pan-evaporation gauges to predict daily black gram evapotranspiration. As noted by the authors, the new stochastic model for black gram evapotranspiration was found to predict daily black gram evapotranspiration with high accuracy ($R^2 = 0.94$). Dynamic stochastic modelling, with a focus on the marginal probability distribution function (known as cumulative distribution function), has been also used for quantifying the PE uncertainty associated with irrigation scheduling [11–13]. Recently, an application of vine copulas with a focus on the short-term structure of the daily evaporation process has been presented [14]. Rainfall-runoff approaches have been presented using stochastic inputs of precipitation and PE to overcome the lack of Penman–Monteith estimates and long-term gauge inputs [15,16].

A substantial amount of previous works have focused on the trend PE assessment [17,18] in conjunction with the well-known term, evaporation paradox [19,20]. The later has been defined as the assumption that, under warming climate and higher temperatures, increased PE rates are expected; however, gauge data show the opposite because observations across the U.S. and the globe show a decreasing trend in pan evaporation. Recent studies recommend the revision of common trend tests through re-evaluation of the statistical significance of an observed trend in a timeseries by assuming a model exhibiting the scaling hypothesis [21], which is shown to be apparent in most key hydrological-cycle processes [22] and provides a more accurate modelling framework than a trend-based approach [23].

The stochastic structure of the PE process, ranging from hourly to climatic scales, is studied here in terms of Hurst–Kolmogorov (HK) dynamics, which describes all processes exhibiting the Hurst phenomenon (i.e., with a power-law autocorrelation function at large scales). Additionally, we focus on the marginal structure of the PE process as fitted through the Pareto–Burr–Feller (PBF) distribution function [24], which includes a large variety of tail-behaviors [25]. Both marginal and second-order dependence structures of the HK dynamics are estimated and compared to the ones identified from global-scale analyses in other key hydrometeorological processes that form the hydrological-cycle path driven by atmospheric turbulence [26], such as temperature, wind, solar radiation, and relative humidity [22,27–29].

Because observations for the PE process are usually found on monthly or daily resolutions, here we use two datasets. The first dataset comprises PET timeseries with monthly resolution extracted from the California Irrigation Management Information System (CIMIS) network in California, comprising 41 ground stations. For the second dataset, we extracted gridded reanalysis PEV data of hourly resolution. In particular, we retrieved the reanalysis data for the grid points in the same area of the network of the ground stations,

so that we could compare its stochastic structures to the PET records, with a focus on the long-range dependence (LRD) behavior.

In Section 2, we introduce the methodology on the estimation of the marginal and second-order dependence structures, while in Section 3, we present the statistical characteristics of the selected stations as well as the results obtained from the analysis, with a focus on the marginal and the dependence structures. Finally, in Sections 4 and 5, we summarize our findings, and we discuss how the results may be consistent with the ones obtained from the hydrological-cycle path under HK dynamics, expanding from Gaussian to Pareto-type tail behavior, and from fractal and intermittent behavior at small scales to LRD behavior at large scales.

## 2. Metrics of Marginal and Dependence Structures

The estimators and models applied for both the marginal and the second-order dependence structures are part of the stochastic framework of the HK dynamics, with a focus on the LRD behavior [30–35], and they have been applied to turbulent and key hydrological-cycle processes of global networks with resolutions spanning from small scales (relevant to the fractal behavior) to climatic scales (for a review, see [26]).

It has been shown that a flexible probability distribution function, which seems to fit well a great variety of key hydrological-cycle processes [25,26], with tail-behaviors ranging from Gaussian to Pareto, is the PBF distribution function [24,36–38], i.e.:

$$F(x) = P\{\underline{x} \le x\} = 1 - \left(1 + \zeta\xi\left(\frac{x-d}{\lambda}\right)^\zeta\right)^{-\frac{1}{\zeta\xi}} \tag{1}$$

where $x > d$, $d$ is a location parameter (in units of $x$), $\zeta$ and $\xi$ are dimensionless shape parameters, and $\lambda$ is a scale parameter (in units of $x$). It is noted that here the Dutch convention is adopted, where underlined symbols denote random variables and stochastic processes.

The estimation of the parameters of the PBF distribution function for the identification of the marginal structure of the PE process is based on the first four statistical moments, and particularly on the central moments and coefficients (i.e., mean, variance, skewness, and kurtosis). It is stressed that, although the estimation from the classical moments of high order are unknowable, especially in the presence of heavy tails and LRD [25], the hourly PEV and the monthly PET processes are expected to be close to a light-tail behavior and, therefore, the estimation of skewness and kurtosis coefficients could be, in approximation, reliably estimated from data.

For the dependence structure of the PE processes, we select the climacogram metric, which is defined as the variance of the averaged process at the scale domain [7]. i.e.:

$$\gamma(k) := \mathrm{Var}\left[\int_0^k \underline{x}(y)\mathrm{d}y\right]/k^2 \tag{2}$$

where $k$ is the scale (in units of $\underline{x}$). (See discussion on the origins of the name, mathematical definitions, etc., in [26,39])

It has been shown that the climacogram estimator at the scale domain is a more powerful estimator than the autocovariance function at the lag domain or the power-spectrum at the frequency domain [34], while its classical estimator adjusted for bias is defined as [40]:

$$\hat{\underline{\gamma}}(\kappa\Delta) = \frac{1}{\lfloor n/\kappa \rfloor}\sum_{i=1}^{\lfloor n/\kappa \rfloor}\left(\underline{x}_i^{(\kappa)} - \hat{\underline{\mu}}\right)^2 + \gamma(\lfloor n/\kappa \rfloor \kappa\Delta) \tag{3}$$

where $\kappa = k/\Delta$ is the dimensionless scale, $\Delta$ is the time resolution of the process, $\hat{\mu}$ is the mean of the process, $\lfloor n/\kappa \rfloor$ is the integer part of $n/\kappa$, and $\underline{x}_i^{(\kappa)}$ is the *i*-th element of the averaged sample of the process at scale $\kappa$, i.e.:

$$\underline{x}_i^{(\kappa)} = \frac{1}{\kappa} \sum_{j=(i-1)\kappa+1}^{i\kappa} \underline{x}_j \tag{4}$$

For the climacogram model, contained in the above estimator, we select a generalization of the HK model (for details and more sophisticated models, see [25,26]), which has been shown to well simulate processes from sub-hourly to over-annual resolutions, and from short- to long-term scales associated with fractal and LRD behaviors that exclude the drop of variance at the intermediate scales:

$$\gamma(k) = \frac{a}{\left(1 + (k/q)^{2M}\right)^{(1-H)/M}} \tag{5}$$

where *a* is the variance of the process, *q* is a scale parameter (in units of the scale *k*), *M* is the fractal parameter, and *H* is the Hurst parameter indicative of the LRD of the process, i.e., for $0.5 < H < 1$ the process exhibits LRD behavior, while for $0 < H < 0.5$ it exhibits an anti-persistent behavior, and for $H = 0.5$ a white-noise behavior. Here, the standardized climacogram is used, i.e., $\hat{\gamma}(k)/\hat{\gamma}(1)$, because the effect of the sample variance is already accounted for through the marginal fitting. We also note that a Gaussian process with $q \to 0$ and $M = 0.5$ coincides with the well-known fractional Gaussian noise model (e.g., [41]).

### 3. Data Extraction and Processing

For the analysis of the hourly PEV process, we use the reanalysis ensemble data extracted (access date at 29/10/2021; with coordinates S32-N42 and W115-E125) from the ERA5 [42] of the Centre for Medium-Range Weather Forecasts (ECMWF; https://cds.climate.copernicus.eu/ accessed on 1 October 2021) across California (Figure 1) and for the period 1979–today (Table 1).

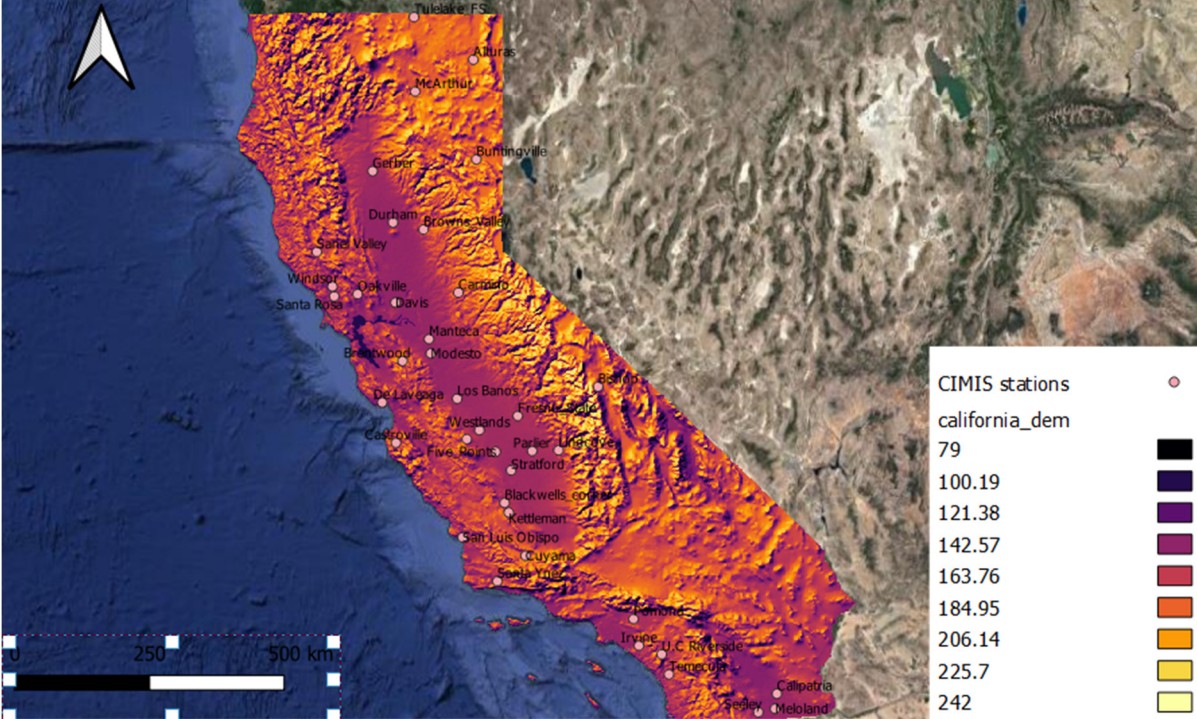

**Figure 1.** Map of Locations of the selected stations.

**Table 1.** Information on the selected stations and the reanalysis data.

| Sequence Number | Name | Process | Temporal Resolution | Time Period | Number of Data Values | Mean (mm) | Standard Deviation (mm) | Skewness Coefficient |
|---|---|---|---|---|---|---|---|---|
| 1 | Five Points | PET | monthly | 1982–2013 | 363 | 131.5 | 73.7 | 0.0 |
| 2 | Davis | PET | monthly | 1982–2013 | 372 | 120.6 | 68.7 | 0.0 |
| 3 | Firebaugh Teles | PET | monthly | 1982–2013 | 370 | 118.1 | 68.9 | 0.1 |
| 4 | Gerber | PET | monthly | 1982–2013 | 370 | 117.3 | 67.9 | 0.1 |
| 5 | Durham | PET | monthly | 1982–2013 | 369 | 107.8 | 61.7 | 0.1 |
| 6 | Carmino | PET | monthly | 1982–2013 | 369 | 116.8 | 68.8 | 0.3 |
| 7 | Stratford | PET | monthly | 1982–2013 | 369 | 128.2 | 75.4 | 0.0 |
| 8 | Castorville | PET | monthly | 1982–2013 | 368 | 79.9 | 32.0 | 0.1 |
| 9 | Kettleman | PET | monthly | 1982–2013 | 368 | 130.4 | 73.9 | 0.0 |
| 10 | Bishop | PET | monthly | 1983–2013 | 363 | 125.5 | 60.9 | 0.0 |
| 11 | Parlier | PET | monthly | 1983–2013 | 362 | 112.5 | 66.0 | 0.1 |
| 12 | Calipatria | PET | monthly | 1983–2013 | 360 | 151.2 | 65.2 | −0.1 |
| 13 | Mc_Arthur | PET | monthly | 1983–2013 | 357 | 101.2 | 66.2 | 0.2 |
| 14 | UC_Riverside | PET | monthly | 1985–2013 | 337 | 121.9 | 47.0 | 0.1 |
| 15 | Brentwood | PET | monthly | 1985–2013 | 327 | 115.8 | 68.1 | 0.1 |
| 16 | San_Luis_Obispo | PET | monthly | 1986–2013 | 327 | 107.5 | 39.5 | −0.1 |
| 17 | Blackwells_corner | PET | monthly | 1987–2013 | 321 | 128.9 | 73.1 | 0.2 |
| 18 | Los Banos | PET | monthly | 1988–2013 | 301 | 119.8 | 70.4 | 0.1 |
| 19 | Buntigville | PET | monthly | 1986–2013 | 325 | 112.9 | 67.8 | 0.1 |
| 20 | Temecula | PET | monthly | 1986–2013 | 320 | 113.5 | 39.9 | 0.0 |
| 21 | Santa_Ynez | PET | monthly | 1986–2013 | 320 | 105.1 | 46.3 | 0.0 |
| 22 | Seeley | PET | monthly | 1987–2013 | 314 | 159.7 | 69.1 | −0.1 |
| 23 | Manteca | PET | monthly | 1987–2013 | 308 | 109.7 | 64.7 | 0.1 |
| 24 | Modesto | PET | monthly | 1987–2013 | 312 | 110.7 | 64.9 | 0.1 |
| 25 | Irvine | PET | monthly | 1987–2013 | 309 | 105.0 | 39.4 | 0.1 |
| 26 | Oakville | PET | monthly | 1989–2013 | 292 | 103.8 | 55.5 | 0.0 |
| 27 | Pomona | PET | monthly | 1989–2013 | 291 | 103.4 | 44.7 | 0.1 |
| 28 | Frenso_State | PET | monthly | 1988–2013 | 297 | 117.7 | 71.2 | 0.1 |
| 29 | Santa_Rosa | PET | monthly | 1990–2013 | 282 | 93.9 | 50.9 | 0.0 |
| 30 | Browns_Valley | PET | monthly | 1989–2013 | 291 | 112.2 | 65.4 | 0.1 |
| 31 | Lindcove | PET | monthly | 1989–2013 | 290 | 110.4 | 65.9 | 0.1 |
| 32 | Meloland | PET | monthly | 1989–2013 | 283 | 153.3 | 66.5 | −0.1 |
| 33 | Alturas | PET | monthly | 1989–2013 | 291 | 97.0 | 60.7 | 0.3 |
| 34 | Cuyama | PET | monthly | 1989–2013 | 289 | 128.4 | 61.4 | 0.1 |
| 35 | Tulelake | PET | monthly | 1990–2013 | 291 | 96.4 | 60.6 | 0.2 |
| 36 | Goleta_foothills * | PET | monthly | 1990–2013 | 197 | 99.1 | 34.8 | 0.0 |
| 37 | Windsor | PET | monthly | 1990–2013 | 266 | 96.4 | 53.6 | 0.1 |
| 38 | De_Laveaga | PET | monthly | 1990–2013 | 274 | 88.6 | 39.4 | −0.1 |
| 39 | Westlands | PET | monthly | 1992–2013 | 255 | 131.2 | 76.0 | 0.0 |
| 40 | Sanel_Valley | PET | monthly | 1990–2013 | 269 | 107.2 | 62.8 | 0.1 |
| 41 | Santa_Monica | PET | monthly | 1993–2013 | 246 | 99.1 | 34.9 | 0.0 |
| 42 | CIMIS (overall) | PET | monthly | 1983–2013 | 12985 | 114.4 | 63.5 | 0.2 |
| 44 | ERA5 | PEV | hourly | 1979–2021 | $0.93 \times 10^6$ | 0.08 | 0.11 | 1.5 |

* There is a large gap in timeseries from 1995–2001.

Observations for the PE processes are usually available in monthly or daily resolutions and usually only for short periods, while a global gridded dataset based on the ERA5 data has been recently released [43]. Here, we use two datasets and compare the marginal and dependence structures of the reanalysis PEV timeseries with the PET timeseries of coarser monthly resolution, extracted from a network of 41 ground stations (see details in Table 1 and Figures 2 and 3). Particularly, the monthly Penman–Monteith dataset of the CIMIS network is used, in which reference evapotranspiration and potential evapotranspiration coincide due to local surface and vegetation conditions. The samples of 41 meteorological stations (https://cimis.water.ca.gov/, accessed on 1 October 2021) are well-distributed across California (Figure 1) for the period 1983–2013 (Table 1), which corresponds to a maximum of 372 monthly values. The meteorological network has been developed in cooperation with Davis University, and the local environment of the meteorological stations allow accurate estimation of the PET.

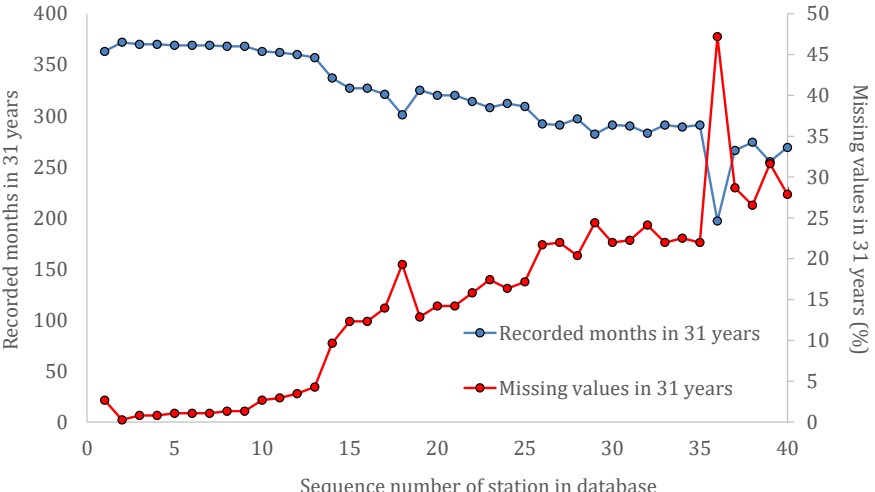

**Figure 2.** Total recorded and missing values of the PET timeseries for each station.

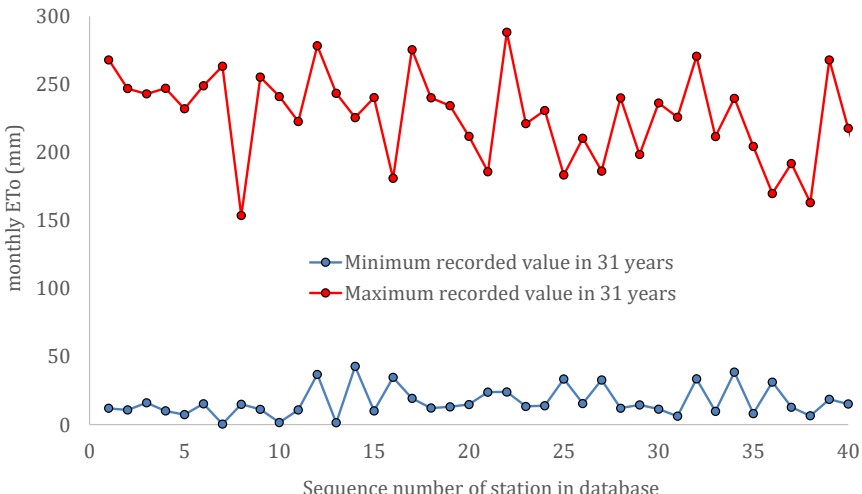

**Figure 3.** Minimum and maximum values of the PET timeseries for each station.

To account for the impact of the double periodicity (diurnal and seasonal) of the PE processes on the dependence structure, we simulate the transformed process by applying a double standardization on the original timeseries. Particularly, we subtract the hourly and monthly means (Figures 4 and 5) and then we divide with the hourly and monthly standard deviations (Figures 6 and 7). Other transformation methods could be applied that take into consideration higher moments (e.g., [26]) such as skewness (Figure 8) and kurtosis

(Figure 9) coefficients, or even more sophisticated ones [44]; however, as can be derived from Table 1 and Figures 6 and 7, the PE processes (especially the aggregated PET process) is close to a light-tail distribution, and therefore we do not expect any significant differences by applying those methods. After the double standardization, we de-standardize each timeseries based on the total mean and standard deviation of the original timeseries (Table 1 and Figure 10). Finally, we fit the marginal and dependence models described in the previous section to each transformed timeseries, and the results are depicted and described in the next section.

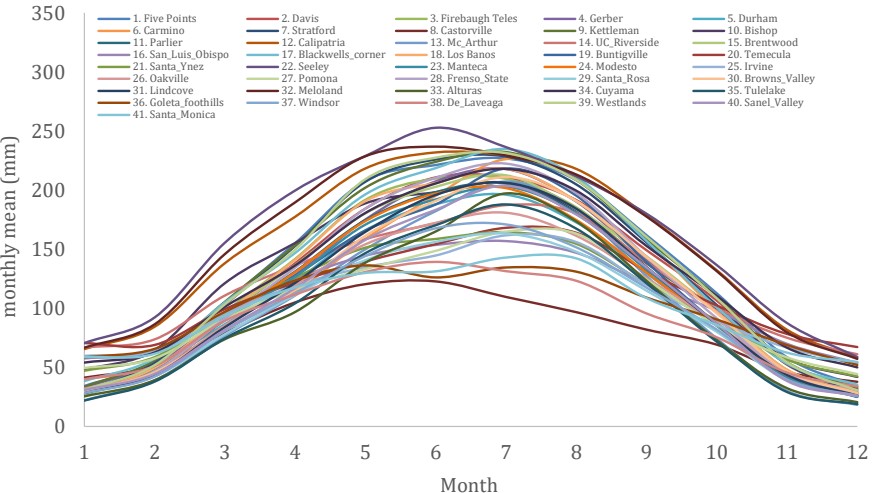

**Figure 4.** Monthly means of the PET timeseries for each station.

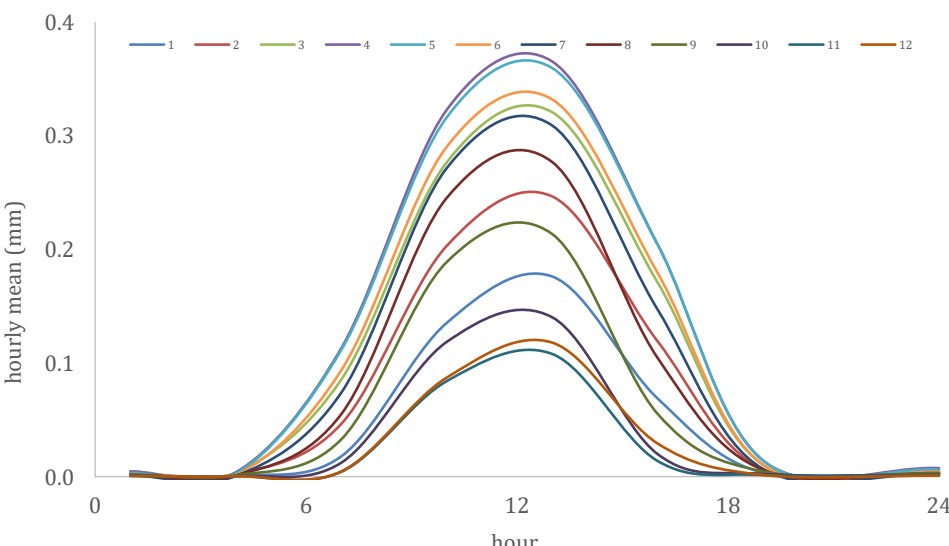

**Figure 5.** Hourly means of the PEV timeseries for each month.

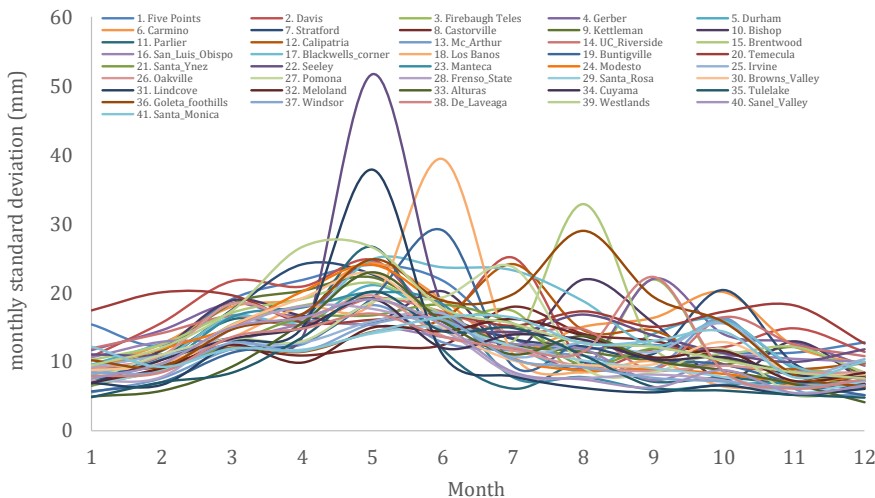

**Figure 6.** Monthly standard deviations of the PET timeseries for each station.

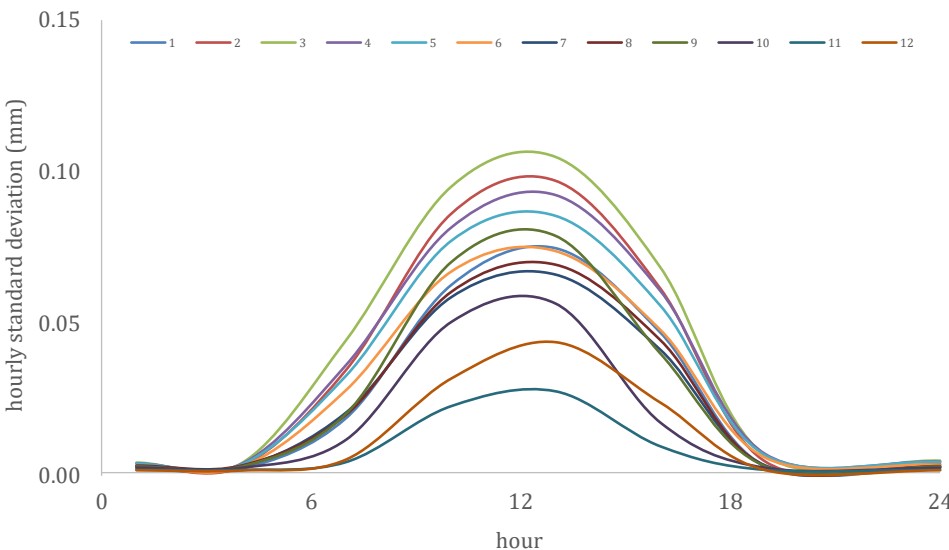

**Figure 7.** Hourly standard deviations of the PEV timeseries for each month.

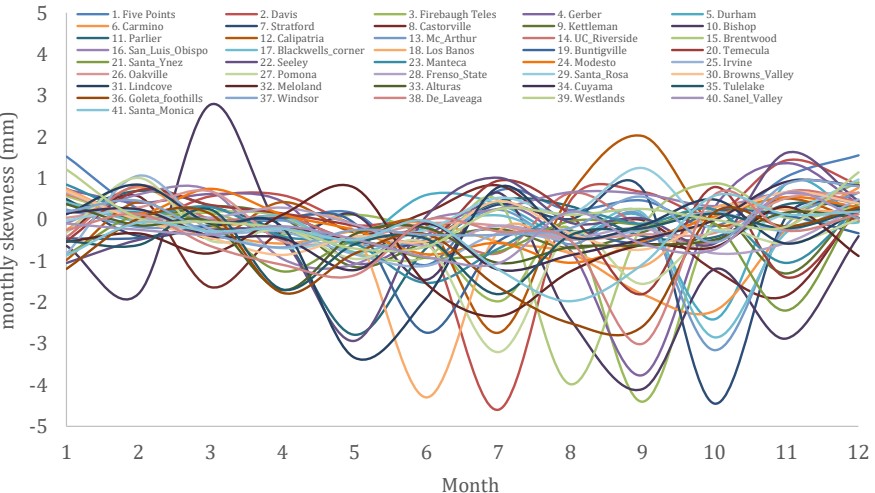

**Figure 8.** Monthly skewness coefficients of the PET timeseries for each station.

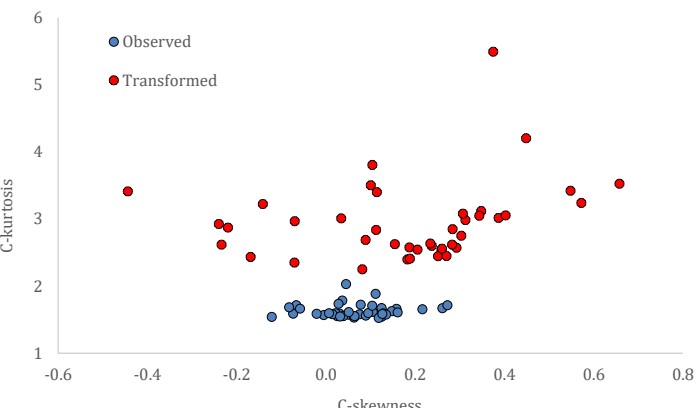

**Figure 9.** Skewness and kurtosis coefficients of the PET timeseries for all stations.

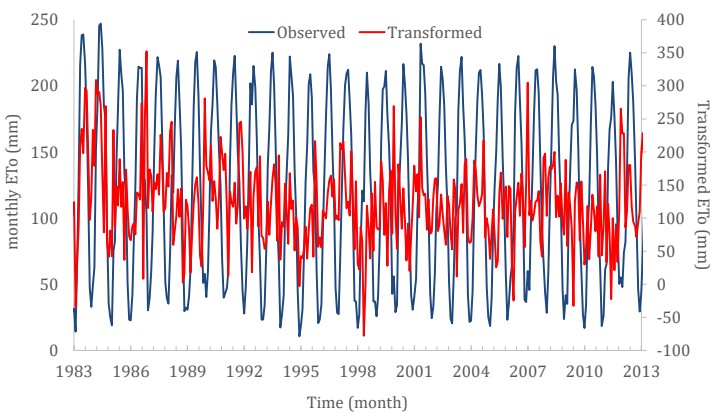

**Figure 10.** Observed and transformed PET timeseries at the Davis station.

## 4. Results

The PBF marginal distribution function is fitted to each transformed timeseries (e.g., Figure 11), and the parameters for all transformed timeseries can be seen in Table 2. Note that the fit of the PBF to all timeseries is exceptionally good. From Table 2, it can be observed that the transformed PEV and PET processes exhibit a light-tail behavior. The average values of the shape parameters are estimated as $\xi \approx 0.04$ and $\zeta \approx 5.7$ for the CIMIS dataset, and $\xi = 0.08$ and $\zeta = 7.6$ for the ERA5 transformed timeseries. It has been shown [25] that the tail index, $\xi$, does not depend on the averaging scale. Therefore, the slight differences in the estimated values are either due to statistical uncertainty or to differences in the nature of the data.

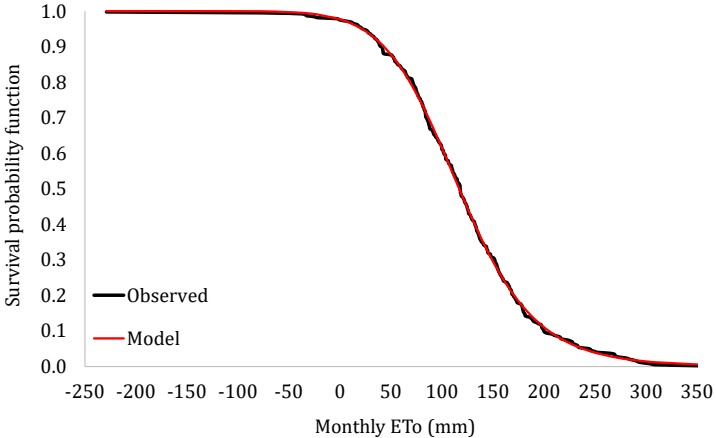

**Figure 11.** Observed and theoretical results of the PBF marginal distribution function of the PET transformed timeseries at the Davis station.

**Table 2.** Parameters of the marginal probability distribution function for all transformed timeseries of each station (note that the squared correlation coefficient is $R^2 > 0.99$ for all models). The symbols $\xi$, $\zeta$, $\lambda$, and $d$ correspond to Equation (1).

| Sequence Number | Name | $\xi$ | $\zeta$ | $\lambda$ (mm) | $d$ (mm) |
|---|---|---|---|---|---|
| 1 | Five Points | 0.100 | 4.5 | 240.0 | −105.0 |
| 2 | Davis | 0.094 | 9.2 | 355.1 | −236.0 |
| 3 | Firebaugh Teles | 0.071 | 8.0 | 353.2 | −227.5 |
| 4 | Gerber | 0.078 | 8.2 | 349.6 | −227.9 |
| 5 | Durham | 0.063 | 6.3 | 285.2 | −167.6 |
| 6 | Carmino | 0.034 | 5.8 | 326.1 | −191.1 |
| 7 | Stratford | 0.054 | 8.0 | 427.1 | −286.5 |
| 8 | Castorville | 0.049 | 5.4 | 132.7 | −45.7 |
| 9 | Kettleman | 0.018 | 4.7 | 308.4 | −154.0 |
| 10 | Bishop | 0.067 | 10.4 | 304.9 | −171.3 |
| 11 | Parlier | 0.042 | 6.4 | 326.8 | −199.4 |
| 12 | Calipatria | 0.093 | 7.7 | 285.2 | −131.2 |
| 13 | Mc_Arthur | 0.038 | 6.8 | 340.7 | −223.3 |
| 14 | UC_Riverside | 0.071 | 4.2 | 145.7 | −14.8 |
| 15 | Brentwood | 0.072 | 7.4 | 344.0 | −221.1 |
| 16 | San_Luis_Obispo | 0.077 | 4.6 | 138.0 | −25.1 |
| 17 | Blackwells_corner | 0.001 | 5.1 | 352.9 | −196.8 |
| 18 | Los Banos | 0.056 | 7.2 | 367.4 | −235.1 |
| 19 | Buntigville | 0.025 | 6.0 | 327.6 | −193.9 |
| 20 | Temecula | 0.074 | 5.1 | 145.7 | −26.0 |
| 21 | Santa_Ynez | 0.012 | 5.1 | 199.9 | −78.5 |
| 22 | Seeley | 0.085 | 7.7 | 285.3 | −116.6 |
| 23 | Manteca | 0.046 | 4.3 | 233.0 | −107.6 |
| 24 | Modesto | 0.013 | 3.7 | 231.3 | −100.0 |
| 25 | Irvine | 0.031 | 4.1 | 132.3 | −15.5 |
| 26 | Oakville | 0.002 | 3.5 | 188.2 | −65.3 |
| 27 | Pomona | 0.025 | 6.0 | 208.3 | −91.1 |
| 28 | Frenso_State | 0.031 | 3.3 | 210.4 | −72.4 |
| 29 | Santa_Rosa | 0.026 | 3.6 | 169.3 | −61.1 |
| 30 | Browns_Valley | 0.004 | 4.4 | 279.5 | −143.7 |
| 31 | Lindcove | 0.045 | 6.2 | 315.7 | −190.4 |
| 32 | Meloland | 0.029 | 5.2 | 268.0 | −94.7 |
| 33 | Alturas | 0.019 | 5.0 | 259.4 | −142.3 |
| 34 | Cuyama | 0.030 | 6.8 | 343.4 | −197.5 |
| 35 | Tulelake | 0.022 | 5.2 | 262.2 | −146.8 |
| 36 | Goleta_foothills | 0.017 | 4.7 | 129.3 | −18.5 |
| 37 | Windsor | 0.001 | 3.1 | 166.1 | −51.4 |
| 38 | De_Laveaga | 0.001 | 5.5 | 195.3 | −90.9 |
| 39 | Westlands | 0.018 | 3.2 | 230.2 | −76.6 |
| 40 | Sanel_Valley | 0.001 | 6.6 | 385.2 | −252.7 |
| 41 | Santa_Monica | 0.016 | 4.7 | 144.3 | −33.4 |
| 42 | CIMIS (meanl) | 0.040 | 5.7 | 260.8 | −132.4 |
| 43 | ERA5-PEV | 0.076 | 7.6 | 0.63 | −0.54 |

Additionally, the combined climacogram from all the empirical ones for the CIMIS transformed timeseries is depicted in Figure 12 and compared to the one from the ERA5 transformed timeseries depicted in Figure 13. It can be observed that a Hurst–Kolmogorov behavior is detected in both data sources, with a Hurst parameter of approximately 0.65. Specifically, the estimated parameters for the CIMIS dataset are $H = 0.64$ and $q = 1.17$ months ($M$ is assumed to be 0.5 because the empirical climacogram is very close to an fGn process), and for the ERA5 timeseries they are $H = 0.69$, $q = 19.7$ h, and $M = 0.8$.

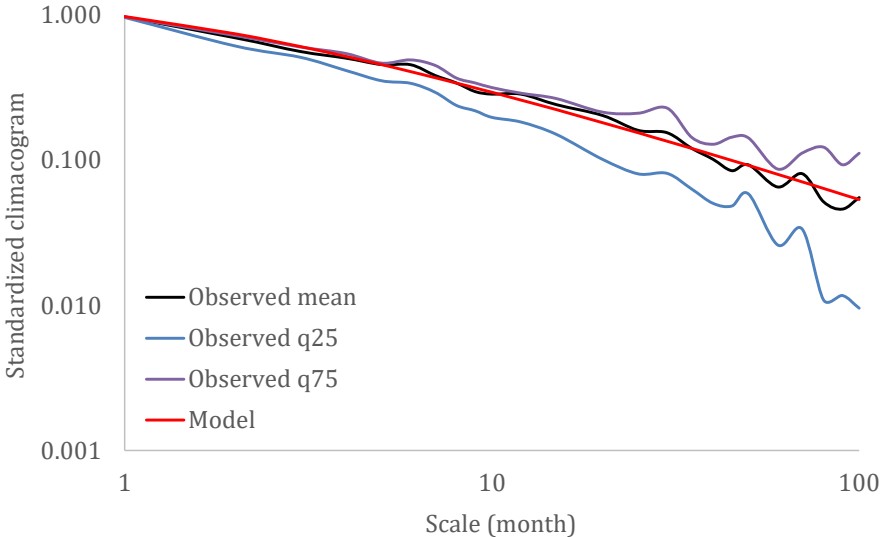

**Figure 12.** Observed and theoretical climacograms through the HK model for all the available PET transformed timeseries adjusted for bias, with the 25% and 75% quantiles (note that the coefficient of determination for the model is $R^2 = 0.993$).

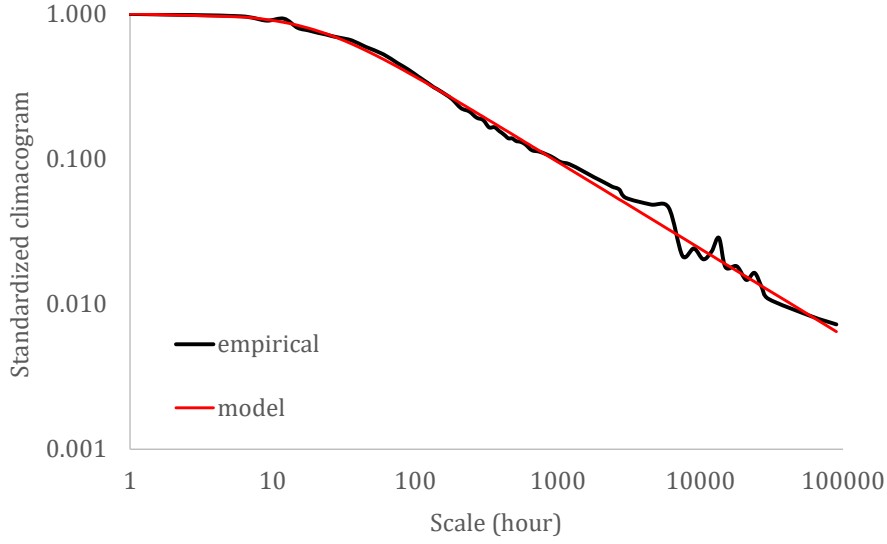

**Figure 13.** Observed and theoretical climacograms through the HK model for the ERA5 transformed timeseries adjusted for bias (note that the coefficient of determination for the model is $R^2 = 0.997$).

## 5. Discussion

Here we discuss how the above results can contribute to the existing literature relating to the potential evaporation and evapotranspiration from the point of view of stochastics and, in particular, of the HK dynamics.

The stochastic analysis of the potential evaporation (PEV) and potential evapotranspiration (PET) presented is useful (a) to highlight the stochastic similarities between them,

(b) to quantify the variability and uncertainty of these processes, and (c) to develop a stochastic model capable of simulating important stochastic characteristics, for purposes such as forecasting and risk management. The PEV timeseries is extracted in hourly resolution as a reanalysis ensemble over California and through the ERA5 network, while for the PET, the high-quality CIMIS dataset with 41 stations is used over the same area for comparison.

The analysis of the above three tasks is performed based on the stochastic metrics and Hurst–Kolmgorov (HK) dynamics. Moreover, the marginal structures and second-order dependence structures are compared to the structures of each other and of other key hydrological-cycle processes such as temperature, relative humidity, wind speed, streamflow, and precipitation, as analyzed from a global network of stations in [25].

In particular, and similar to the global analysis, it is illustrated how the Pareto–Burr–Feller (PBF) probability distribution function may well describe the marginal structure of both the hourly PEV and monthly PET. Additionally, both processes are shown to exhibit a light-tail behavior. However, it is noted that the shape parameters of the PBF (i.e., $\xi$ and $\zeta$), which characterize the type of the tail, are slightly smaller in the CIMIS data (i.e., overall mean from stations 0.04 and 5.7, respectively) as compared to the reanalysis data (i.e., 0.08 and 7.6, respectively), indicating a heavier tail for the latter.

Additionally, it is found that, similarly to the other key hydrological-cycle processes mentioned above, both PEV and PET processes exhibit long-range dependence, with a Hurst parameter of medium strength. In particularly, *H* is estimated as 0.65 and 0.68 for the PET and PEV processes, respectively, which is weaker than the ones for temperature, relative humidity, solar radiation, and wind speed (0.80–0.85 [25]) and stronger than the one for precipitation (i.e., 0.61 [25]) for the examined range of scales spanning from the hourly resolution to the climatic scales. This can be interpreted as an indication that the PET and PEV processes have a wider predictability time window than precipitation's, and narrower than the rest (i.e., entailing a higher degree of long-term unpredictability).

As a final remark, the need to apply a suitable stochastic model to reproduce important characteristics, such as LRD behavior, is stressed. The work shows the robust use of a stochastic framework to simulate the variability and uncertainty of a hydrometeorological process in emerging new practices and challenges:

- Stochastic modelling of evapotranspiration at a fine time scale (e.g., hourly) is considered to be useful for numerous agronomist applications because it is strongly connected to the forecast of the plant water demands. In recent years of micro-farm techniques, the stochastic modelling of evapotranspiration, with sound physical-interpretation, has tracked the attention of the scientific community in order to simulate more accurately the water-food-energy nexus.
- A proper stochastic model for the simulation of the evapotranspiration should be based at a wide range of spatio-temporal scales and meteorological conditions; thus, a global-scale analysis is important in order to identify stochastic similarities so as to improve the simulation techniques.
- Stochastic simulation of the error analysis between the modelled and the measured Penman–Monteith assessment could highly contribute to improving potential evapotranspiration estimates.
- Stochastic PET modeling could offer a solid probabilistic frame for identifying the long-term trend of hydrometeorological components in horizons greater than the available records and thus is of potential interest for climatological studies.

## 6. Conclusions

A stochastic model is presented for hourly potential evaporation (PEV) and monthly potential evapotranspiration (PET) based on the ERA5 hourly reanalysis data and the Penman–Monteith model applied to the well-known CIMIS network.

It was found that both the marginal probability distributions of PEV and PET are light-tailed when estimated through the Pareto–Burr–Feller distribution function. Additionally,

the long-range dependence of both the PEV and PET is found to be of moderate strength, quantified through a Hurst parameter of 0.64 and 0.69, respectively.

The above results reveal the stochastic similarities between the ground and reanalysis data series. Additionally, the results are shown to be consistent to the hydrological-path of the marginal and dependence structures of Hurst–Kolmogorov dynamics. In particular, both PET and PEV can be placed between the stochastic structures of temperature, relative humidity, solar radiation, and wind speed (i.e., strong LRD and light- to medium-tail) and the precipitation's structures (i.e., weak LRD and heavy tail). Finally, it is discussed how the existence of, even moderate, long-range dependence and tail distribution increase the variability and uncertainty of both processes, and thus limit their predictability.

**Author Contributions:** Conceptualization, P.D.; methodology, P.D. and D.K.; formal analysis, P.D.; investigation, P.D. and A.T.; data curation, A.T.; writing—review and editing, D.K. All authors have read and agreed to the published version of the manuscript.

**Funding:** This research received no external funding.

**Informed Consent Statement:** Informed consent was obtained from all subjects involved in the study.

**Data Availability Statement:** Hersbach, H. et al., (2018) was downloaded from the Copernicus Climate Change Service (C3S) Climate Data Store. The results contain modified Copernicus Climate Change Service information 2020. Neither the European Commission nor ECMWF is responsible for any use that may be made of the Copernicus information or data it contains.

**Acknowledgments:** We thank both the anonymous reviewers for their suggestions and comments, and the editors for handling the manuscript.

**Conflicts of Interest:** The authors declare no conflict of interest.

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
