# Peer review of "Stochastic Analysis of Hourly to Monthly Potential Evapotranspiration with a Focus on the Long-Range Dependence and Application with Reanalysis and Ground-Station Data"

_hydrology, doi:10.3390/hydrology8040177_

Round 1
Reviewer 1 Report
Dimitriadis et al. (2021) have performed a stochastic analysis applied to evaporation and evapotranspiration data derived from both ground stations and reanalysis. This thematic is within the scope of Hydrology and would be interesting to the readers of the journal. In my opinion, the manuscript, as it is now, needs several clarifications. The major flaw is the lack of a proper discussion section, since there is no comparison of the results obtained with previously published papers. Please see the comments below.
Lines 24-28: Please define ET, ETo and PE in this paragraph.
Lines 33-36: Not clear the advantages of using a stochastic model. It does not need meteorological inputs? Please better explain this model and highlight its major advantages.
Line 38: Why is it ubiquitous?
Lines 38-40: Not clear so far what is a stochastic model. Are the ARIMA and Winters’ exponential smoothing models examples of stochastic models?
Line 38: Please add the R2 or the statistical metric used by the authors to define the high accuracy of the model.
Line 48: Please define the “marginal distribution function”.
Lines 49-51: Please discuss the results of this citation.
Line 55: Please define the term “term evaporation paradox”.
Lines 55-58: Confusing sentence. Please rewrite it.
Line 60: Please define the term “Hurst-Kolmogorov (HK) dynamics”.
Line 69: Where are these land stations located?
Line 71: Ground stations or land stations? Please standardize this term in the entire manuscript.
Lines 130-134: Please better discuss the ERA5 dataset in here (e.g., spatial and temporal resolution, examples of works that used this dataset). Moreover, why did you choose California as your study area?
Line 147: This is a very poor figure. All cartographical elements are missing (e.g., scale, north arrow, legend, coordinates). Please add the source of the base image and when this image was captured. As it is, it seems a screenshot of Google Earth.
Line 148: Source of station data?
Line 150: Why are there so many missing values in station 36? Moreover, clarify in the legend that the maximum number of recorded months in 31 years is 372.
Line 150 and Line 154: Please consider using the name of each station instead of the number.
Line 170, Line 176, and Line 182: Too much information in these figures. Please consider dividing them into smaller figures (e.g., 8 graphs).
Line 173 and Line 179: Is this the average of all stations?
Line 188: Why did you chose this specific station to be represented in this figure?
Line 196: What did you mean by heavier?
Line 208: Please add the description of each symbol in the legend of this table.
Lines 212-213: Please improve this legend. All stations are considered in here?
Lines 216-217: This figure is not mentioned in the text.
Lines 225-242: This seems like an abstract of the results. No discussion of them made so far.
Lines 243-246: This paragraph was shown in the results section.
Line 264: No comparison of your results with previously published papers.
Line 264: Please divide this section into Discussions and Conclusions (although there are not discussions of your results in here). It is an easier reading.
Author Response
Please find attached our detailed responses for addressing reviewer's comments

Reviewer 2 Report
Dear editor,
In this work, the authors present a stochastic simulation of the error analysis between modelled ERA5 data [40] versus the measured Penman-Monteith evapotranspiration [monthly PET(ΕΤο) data from the CIMIS network].
The submitted work somehow comes to answer the comment that is reported in a recent work of Singer et al., 2021 [40]: "there is currently no global set of PET data developed with conventional models, such as the method Penman-Monteith, in recent decades with combined high spatial and time requirements input elements of many environmental models." Thus, the submitted work acquires tremendous interest and considerable originality.
The authors used data from the CIMIS network of ground meteorological stations. This extensive network is considered the most reliable globally on Reference evapotranspiration (ETo/ETp) estimates (high quality & long time series). Following analysis, the authors in section 2 (Metrics of Marginal and Dependence Structures) adopt a stochastic approach that shows how much the uncertainty of both processes could increase.
However, the essential elements of the Penman-Monteith method (and differs from the original Penman model) are that it introduces the bulk surface resistance of the crop canopy and soil (Crop surface resistance (rs), aerodynamic resistance (ra) and the modified Psychometric constant (γ*). The P-M equation is a single-layer model where the resistances for vegetation and soil are assumed to reside in parallel.
Recommendations:
- In my opinion, the term "hourly to monthly evaporation" should be replaced with "hourly to monthly, Potential evapotranspiration". In this case (for CIMIS stations according to the grass reference, ETo, PET, ETa is identical. see definitions 1,2,3 and more references).
- Line 12. Strongly recommended replacing "sunshine" with "Incoming shortwave radiation" and "humidity" with "Relative humidity."
- There is no any comment for figure 11 in the text.
-----------------------
CIMIS data processing involves checking the accuracy of the measured weather data for quality, calculating reference evapotranspiration. The ideal site for a CIMIS weather station is about 8 hectares or more extensive cool-season grass that is well maintained well-watered (especially under the warm season). (https://cimis.water.ca.gov/Stations.aspx)
- ETo: Reference evapotranspiration: "the rate of evapotranspiration from a hypothetical reference crop with an assumed crop height of 0.12 m, a fixed surface resistance of 70 sec/m and an albedo of 0.23, closely resembling the evapotranspiration from an extensive surface of green (cool season) grass of uniform height, actively growing, well-watered, and completely shading the ground"
- PET: The potential evapotranspiration concept was first introduced in the late 1940s and 50s by Penman and it is defined as “the amount of water transpired in a given time by a short green crop, completely shading the ground, of uniform height and with adequate water status in the soil profile”. Note that in the definition of potential evapotranspiration, the evapotranspiration rate is not related to a specific crop.
- ETa: Actual evapotranspirations: the amount of water that is actually removed from a surface due to the processes of evaporation or/and transpiration, under the condition of existing water supply.
Alexandris, S., & Proutsos, N. (2020). How significant is the effect of the surface characteristics on the Reference Evapotranspiration estimates?. Agricultural Water Management, 237, 106181.
Doorenboos, J., Pruitt, W.O., 1977. Guidelines for predicting crop water requirements. FAO Irrigation and Drainage Paper No.24. Food and Agricultural Organization of the United Nations, Rome.
Droogers, P., Allen, R.G., 2002. Estimating reference evapotranspiration under inaccurate data conditions. Irrigation and Drainage Systems 16, 33–45.
Eching, S., Frame, K., Snyder, L., 2002. Role of technology in irrigation advisory services: The CIMIS experience. 18th Congress and 53rd IEC meeting of the International
Temesgen, B., Allen, R.G., Jensen, D.T., 1999. Adjusting temperature parameters to reflect well-watered conditions. Journal of Irrigation and Drainage Engineering 125 (1), 26–33.
Author Response
Dear Editors, attached are the responses with details on the revision of our work

Round 2
Reviewer 1 Report
I appreciate authors’ effort in responding to all my questions. They have addressed to most of them properly. I still consider Figure 1 needs to be improved before publication.